# Genome-Wide Identification of Aquaporin Genes in Adzuki Bean (*Vigna angularis*) and Expression Analysis under Drought Stress

**DOI:** 10.3390/ijms232416189

**Published:** 2022-12-19

**Authors:** Rupesh Tayade, Varnika Rana, Mohammad Shafiqul, Rizwana Begum Syed Nabi, Gaurav Raturi, Hena Dhar, Vandana Thakral, Yoonha Kim

**Affiliations:** 1Department of Applied Biosciences, Kyungpook National University, Daegu 41566, Republic of Korea; 2National Agri-Food Biotechnology Institute (NABI), Mohali 140306, India; 3Department of Southern Area Crop Science, National Institute of Crop Science, Rural Development Administration, Miryang 50424, Republic of Korea

**Keywords:** aquaporin gene, drought, adzuki bean, genome-wide identification, gene structure, phylogeny

## Abstract

The adzuki bean *Vigna angularis* (*Wild.*) is an important leguminous crop cultivated mainly for food purposes in Asian countries; it represents a source of carbohydrates, digestible proteins, minerals, and vitamins. Aquaporins (AQPs) are crucial membrane proteins involved in the transmembrane diffusion of water and small solutes in all living organisms, including plants. In this study, we used the whole genome sequence of the adzuki bean for in silico analysis to comprehensively identify 40 *Vigna angularis* aquaporin (VaAQP) genes and reveal how these plants react to drought stress. VaAQPs were compared with AQPs from other closely-related leguminous plants, and the results showed that mustard (*Brassica rapa*) (59), barrel medic (*Medicago truncatula*) (46), soybean (*Glycine max*) (66), and common bean (*Phaseolus vulgaris* L.) (41) had more AQP genes. Phylogenetic analysis revealed that forty VaAQPs belong to five subfamilies, with the *VaPIPs* (fifteen) subfamily the largest, followed by the *VaNIPs* (ten), *VaTIPs* (ten), *VaSIPs* (three), and *VaXIPs* (two) subfamilies. Furthermore, all AQP subcellular locations were found at the plasma membrane, and intron–exon analysis revealed a relationship between the intron number and gene expression, duplication, evolution, and diversity. Among the six motifs identified, motifs one, two, five, and six were prevalent in VaTIP, VaNIP, VaPIP, and VaXIP, while motifs one, three, and four were not observed in VaPIP1-3 and VaPIP1-4. Under drought stress, two of the VaAQPs (*VaPIP2-1* and *VaPIP2-5*) showed significantly higher expression in the root tissue while the other two genes (*VaPIP1-1* and *VaPIP1-7*) displayed variable expression in leaf tissue. This finding revealed that the selected VaAQPs might have unique molecular functions linked with the uptake of water under drought stress or in the exertion of osmoregulation to transport particular substrates rather than water to protect plants from drought. This study presents the first thorough investigation of VaAQPs in adzuki beans, and it reveals the transport mechanisms and related physiological processes that may be utilized for the development of drought-tolerant adzuki bean cultivars.

## 1. Introduction

Water is important for plant survival and intrinsic physiological functions, and it is also vital for solubilizing nutrients and other essential elements in the soil. Generally, plant roots absorb water and solutes from the soil via different pathways, such as apoplast, symplast, and transcellular pathways, and transmit them to various plant organs via vascular cells [1,2]. Aquaporins (AQPs), a group of membrane proteins found in nearly all living things, including plants, are known to be crucial for the transfer of solutes and water along the transcellular and apoplast pathways [1,3,4]. Animal genomes include fewer AQP genes than plant genomes. For example, plant species such as canola have 120 AQPs, whereas the human genome has only 13 AQPs [5,6]. Plant AQPs have evolved into several subfamilies and groups with varying solute specificities and are classified into five major subfamilies: small basic intrinsic proteins (SIPs), tonoplast intrinsic proteins (TIPs), plasma membrane intrinsic proteins (PIPs), nodulin 26-like intrinsic proteins (NIPs), and unidentified X-intrinsic proteins (XIPs) [7,8].

Adzuki bean (*Vigna angularis*) is a leguminous crop cultivated in different parts of the world; it is a source of high-starch, digestible protein, and essential nutrients [9,10,11]. Adzuki bean has proteins with a balanced diversity of amino acids, resistant starch, active compounds with medicinal values, and high lysine content, making it a very nutritionally- and economically-important crop. Furthermore, the adzuki bean is used in a variety of foods, such as cakes, desserts, pastries, porridge, adzuki rice, jelly, ice cream, and adzuki milk [10,12]. Previous studies have suggested that droughts have an impact on the cultivation of adzuki beans and shown that these plants are vulnerable to drought stress in the early stages of growth as well as altered yield in later stages [13,14,15]. During drought stress, adzuki beans have been shown to limit root and shoot growth as well as physiological and biochemical parameters at the seedling stage [15]. Previous studies have shown that AQPs are promising candidates for plant abiotic stress tolerance and that the opening and closing of cellular gates are controlled by AQPs [16,17,18]. In numerous crops, AQP expression has also been linked to drought and salinity tolerance [18,19,20,21].

Considering the economic, nutritional and health importance of the adzuki bean, efforts have been made toward whole-genome sequencing and genomic resource development [9,22]. Recent rapid development in sequencing tools have facilitated the construction of annotated plant genome sequences and generated enormous amounts of transcriptomic and proteomics data in the public domain. Additionally, the ready accessibility of omics resources has improved our comprehension of the AQP transport mechanism in plants. For instance, genome-wide identification of AQP genes has been performed in numerous plant species, including rice (*Oryza sativa*), soybean (*Glycine max* L.), and bottle gourds (*Lagenaria siceraria*) [23,24,25], and these studies have shown that AQP genes are conserved across plant species. However, AQPs remain poorly understood in many species. For example, in adzuki, AQP gene information is sparse, particularly in terms of drought stress. Hence, to completely comprehend the complex AQP system in plants, researchers should focus on AQP gene regulation and interactions with other proteins and determine the impact of environmental stress conditions on AQP gene functionality and regulation. However, AQP gene expression has not been previously studied in adzuki beans despite the availability of modern genomic technologies, the importance of AQPs in plant–water relations, and the lack of understanding of adzuki responses to drought stress. To gain insights and improve the drought stress tolerance of this important crop, we conducted genome-wide in silico identification of AQP genes using the available genome sequence of adzuki bean and sequences of known AQPs from other plant species (Arabidopsis (*Arabidopsis thaliana*), rice, and soybean), and investigated the phylogenetic relation, chromosome distribution, conserved domain, and protein–protein interactions. Furthermore, using publicly available transcriptome data, we explored the expression patterns of AQPs to identify their functions under abiotic stresses. Generally, wild accessions are considered to have native resistant genes or alleles for tolerance to biotic and abiotic stress with higher genetic diversity. Thus, wild adzuki bean accessions were used to test the relationship between the differential expression of these genes and differential responses to drought stress. The transcript profiles of selected AQP genes were studied in the root and leaf tissues of control and drought stress plants. It is considered that plants show different mechanisms to mitigate drought stress, and evaluation of phenotypic traits related to physiology, such as morphological and biochemical traits, are crucial under abiotic stress. In addition, these traits are altered by various factors and influences (e.g., plant height, chlorophyll content, and soil measure) and are essential for crop improvement; thus, we also investigated phenotypic traits in this study. Mainly, we aim to emphasize the significance of AQP genes in adzuki bean by identifying and validating the AQPs genes and provide the candidate genes which are induced by drought stress for more in-depth functional characterization. This study is the first to thoroughly examine AQPs in adzuki beans. The findings of this work may spur further investigation into the molecular basis of AQPs, which may enable the selection of candidate genes for enhancing drought tolerance in adzuki beans.

## 2. Results

### 2.1. Genome-Wide Distribution and Classification of V. angularis AQPs

The AQP sequences from three distinct plant species were used as a query in a BLASTp (with HSPs) search, and 40 putative AQPs were identified in the *V. angularis* genome. The set of *V. angularis* AQPs included fifteen PIPs, ten NIPs, ten TIPs, three SIPs, and two XIPs (Appendix A). The AQPs were not uniformly distributed in the *V. angularis* genome (Figure 1). Several AQPs are present in clusters on different chromosomes. Seven VaAQP genes were found on chromosome 3, six VaAQP genes were found on chromosome 8, four VaAQP genes were found on chromosomes 2, 5, and 10, three genes were found on chromosome 7, and two VaAQP genes were found on chromosomes 1, 9, and 11. Whereas, single VaAQP genes were found on chromosome 6 and the scaffold as shown in Figure 1. The conserved domain and transporter superfamily of AQPs were confirmed using the CDD tool from the NCBI, which confirms that all VaAQPs belong to the MIP superfamily and most of them contain Asn-Pro-Ala (NPA) signature motifs. Similarly, a typical six-transmembrane domain architecture was observed in all identified VaAQPs (Appendix A), and it showed a homology-based protein structure. However, TMHMM predicted six transmembrane domains in only 27 out of the 40 VaAQPs shown in Table 1.

### 2.2. Phylogenetic Distribution of AQPs in V. angularis

We constructed a phylogenetic tree to reveal the evolutionary interrelatedness of the VaAQP proteins, and it was created based on multiple alignments of full-length sequences of each protein. The 40 VaAQPs were divided into five subfamilies: PIPs, TIPs, NIPs, SIPs, and XIPs (Figure 2A). The largest group was PIP, with 15 members, and the smallest group was XIP, with two members. VaPIP formed two distinct groups, VaPIP1 and VaPIP2, which had eight members and seven members, respectively.

To understand the evolution of AQPs, AQP genes from arabidopsis, rice, and soybean were compared to those from *V. angularis*. The phylogenetic tree of VaAQPs and those of the other three species formed five distinct subfamilies (Figure 2B). The naming of *V. angularis* AQPs was performed based on their grouping with known AQPs from the other three species. Furthermore, *V. angularis* was categorized into fifteen VaPIPs, ten VaNIPs, ten VaTIPs, three VaSIPs, and two VaXIPs. VaNIP1 had five members, while VaNIP2, VaNIP3, and VaNIP4 had one, three, and one member, respectively. Similarly, VaTIPs formed two subgroups, with one consisting of a single subgroup (*VaTIP5-1*) along with three other members of different species (arabidopsis, rice, and soybean) and the other consisting of the subgroups VaTIP1 (*VaTIP1-1*, *VaTIP1-2*, *VaTIP1-3*), VaTIP2 (*VaTIP2-1*, *VaTIP2-2*, *VaTIP2-3*), VaTIP3 (*VaTIP3-1*, *VaTIP3-2*), and VaTIP4-1. This finding shows that the adzuki bean genome has similar subfamily numbers as the TIP AQPs of arabidopsis, rice, and soybean, as previously reported. VaSIPs formed two subgroups (Figure 2B), with VaSIP2 having a single member along with two GmSIP2s (*GmSIP2-1* and *GmSIP2-2*), AtSIP2-1, and *OsSIP2-1* from soybean, arabidopsis, and rice, respectively. VaXIPs form a single group that included *VaXIP1-1* and *VaXIP2-1*.

### 2.3. Gene Structure, Organization, and Evolution of AQPs

To explore the structure of VaAQPs and their evolutionary relationships, we used the MEME suite software tool to perform a gene sequence analysis of the conserved motifs, and it identified six different conserved sequence motifs (motifs 1–6) in the VaAQPs. Interestingly, motif 2 was widely distributed among most VaAQP members except for VaNIP3-3 and VaNIP1-5 (Figure 3A). In VaTIPs, five motifs were observed VaTIP2-1, VaTIP4-1, and VaTIP5-1, while four motifs (motifs 1, 2, 5, and 6) were found in VaTIP1s and VaTIP3s. Similarly, five motifs were found in most VaPIPs except for VaPIP1-3 and VaPIP1-4 (Figure 3A). VaSIP1-1 and VaSIP2-1 had two motifs (motifs 2 and 5), whereas VaSIP1-2 had three motifs. VaXIP1-1 had five motifs, and VaXIP2-1 had four motifs. Overall, the VaAQP motif analysis revealed similar compositions of closely related VaAQPs discovered in phylogenetic clades, implying that VaAQPs belonging to the same phylogenetic group may have similar functions. We further analyzed the gene structure of all identified candidate VaAQPs by an exon–intron organization that showed large variations in the number and length of the introns and exons of VaAQPs, and these variations led to gene length variations (Figure 3B).

### 2.4. Characterization of NPA Motifs, Transmembrane Domains, and Subcellular Localization of Adzuki Bean AQPs

The presence of conserved domains is known to be a signature feature of AQPs, and it can be detected with the help of the NCBI CDD search tool, as detailed in Appendix A. Variations in the NPA domains were observed in different AQPs. Of the ten NIPs, only four NIPs contained both NPA domains, whereas in the other NIPs, the NPA domains were replaced with NPV in loop E (LE) of VaNIP1-3, VaNIP 3-1, VaNIP 3-2, and VaNIP 3-3, and these domains were replaced with NPG and NPS in loop B of VaNIP1-4 and VaNIP3-1, respectively. Similarly, NPA was replaced with NPT in loop B of VaSIP1 and with NPL in VaSIP2. The NPA of VaXIP1-1 and VaXIP2-1 had NPK and SVP in the B loop (Table 2). According to the results generated by TMHMM and SOSUI, the number of transmembrane helices in the VaAQPs can vary from two to eight transmembrane (TM) domains; however, most AQPs contained six TM domains. The CELLO subcellular localization prediction tool localized all AQPs to the plasma membrane, while the WoLF PSORT tool indicated that the subcellular localization may vary (Table 1).

### 2.5. Tertiary Structure of Aquaporin

The precise structure of aquaporin is very important because it provides insights into the activity and interaction of the protein. The 3D structure of a protein can be determined directly using the protein sequence based on various bioinformatics tools. Homology-based 3D structural modeling was performed using Phyre2 and visualized using the MOLE online server, which revealed a typical hourglass-like structure of the AQP protein (Figure 4) and the presence of the six transmembrane domains. The pore profile of VaNIP2-1 showed that the pore length was 63.8 Å and the radius was 2.4 (Figure 4B).

### 2.6. Effect of Drought Stress on Plant Attributes and Soil Moisture Content

To investigate the impact of drought stress on plant attributes such as plant root, shoot, fresh weight, dry weight, and plant height evaluated for control and drought stress plants, all these attribute changes on the 10th day of drought stress are presented in Figure 5A. The drought-stressed plant showed a significant reduction at (*p* < 0.001) level in roots, shoot, fresh weight, and dry weight, as well as plant height (Figure 5). In addition, the total chlorophyll content of drought-stressed plants was slightly higher than the control plants but it was not statistically significant (Figure 5G). We also measured the soil moisture content on the 10th day of drought stress; it was significantly lower in the case of drought-exposed plants compared to the control plants (Figure 5H). Overall, these results suggested that drought stress negatively impacted measured plant attributes as well as reduced soil moisture content.

### 2.7. Expression Profiling of AQPs in V. angularis

To determine the expression of AQPs in different tissues and associated changes under abiotic stress, publicly accessible transcriptome datasets were mined. The transcriptome data revealed that all VaAQPs identified in this study were represented in the dataset. All 40 VaAQPs showed differential expressions towards drought and saline stress (Figure 6). The expression of VaPIP1-6, VaPIP1-7, PIP2-1, PIP2-2, XIP1-1, XIP2-1, TIP3-1, and TIP3-2 increased during drought stress in the 17,235 variety. VaPIP1-2 and VaTIP5-1 were upregulated only in the leaf tissue. In the case of VaSIP1-2, high expression was observed under control conditions, and downregulated expression was observed under stress conditions (Figure 6).

The role of PIP AQP subfamily is well established for managing the water status of plants through controlling cell and tissue hydraulics and is also considered the most likely candidate for protein-mediated hydraulic conductivity in roots and leaves. In addition, based on the transcriptome data mining, we found higher expression of PIP genes in plant tissues (i.e., root, stem and leaves) therefore, we selected the four genes *VaPIP1-1*, *VaPIP1-7*, *VaPIP2-1*, and *VaPIP2-5* for validation and reconfirmation through qRT-PCR. The results showed significantly higher transcript levels of *VaPIP1-1* and *VaPIP1-7* in the control plants in both the root and leaf tissues compared with that in the drought plants. However, under drought stress, *VaPIP2-1* and *VaPIP2-5* showed higher expression in the root tissue but significantly lower expression in the leaf tissue compared with that in the control plants. The results showed that varietal gene differences occurred in the expression trends according to the transcriptome results (Figure 7), with upregulated expression observed for *VaPIP2-1* and *VaPIP2-5*, particularly in roots, during drought stress. All genes were downregulated in the leaf tissue during drought stress. The expression results of selected *VaPIP* genes may help develop a deeper understanding of the roles of *VaPIPs* in coping with drought stress.

## 3. Discussion

In plants, AQPs are essential channel proteins, ubiquitously important for plant health, that are involved in the transport of water and small molecules in and out of biological membranes. Thus, the functions of AQPs allow for improving the resilience of plants to drought, cold, salt, and other environmental stressors. Plants contain large amounts of AQPs [26], and plant genome sequencing is becoming a subject of intensive research. In addition to facilitating the identification and characterization of AQPs in plants, such research has broadened our understanding of their molecular roles and evolutionary diversity. Many AQP genes have been identified in plant species, thus emphasizing their role in water movement, flowering, and drought tolerance. Considering the potential of AQPs, the present study utilized the whole-genome sequence of *V. angularis* to characterize *VaAQPs* and provide insights into how they respond to drought stress. Through in silico analysis, a total of 40 putatively functional AQP genes were identified in the adzuki bean, and they phylogenetically belong to five subfamilies: *VaPIP* (fifteen), the largest group, followed by *VaNIP* (ten), *VaTIP* (ten), *VaSIP* (three), and *VaXIP* (two). These findings are similar to the results obtained for other plant species. Further comparison of the *VaAQPs* results with those from closely related leguminous plants revealed that a slightly higher number of AQP genes was found in Chinese cabbage (*B. rapa* ssp. *pekinensis*) (59) [27], whereas 37 were found in bay bean (*Canavalia rosea*) [28], 40 were found in chickpea (*Cicer arietinum* L.) [29], 46 were found in barrel medic (*Medicago truncatula*) [30], 66 were found in soybean [31] and 41 were found in common bean (*Phaseolus vulgaris* L.) [32]. Among non-leguminous plants, 35 AQPs were found in arabidopsis, 33 were found in rice [33] and 47 were found in tomato (*Solanum lycopersicum*) [34] which are all higher than that found in *V. angularis*. The phylogenetic relatedness of *VaAQPs* among arabidopsis, rice, and soybean showed that among the PIP, TIP, NIP, SIP, and XIP subfamilies, thirteen, ten, nine, three, and none were found in arabidopsis, eleven, ten, eleven, two, and none were found in rice, and twenty-two, twenty-three, seventeen, eight, and two were found in soybean, respectively, suggesting that soybean has the highest PIPs, followed by *V. angularis*. The TIP family shares equal numbers between arabidopsis and rice. Although most plant species do not possess the XIP subfamily, a few plants, such as the common bean, soybean, and bay bean, have been reported to contain members of the XIP subfamily. However, plant crops, such as chickpea and mustard, lack the XIP subfamily. Thus, XIPs may not be involved in significant physiological processes in many plant species. Regarding the localization of AQPs in plants, *Maurel* et al. (*2015*) found that PIP, NIP, and XIP reside on the plasma membrane, while typical TIP and SIP are present in the tonoplast and endoplasmic reticulum (ER), respectively. For *VaAQPs*, based on the CELLO subcellular localization prediction, all AQPs were located in the plasma membrane, as CELLO predictions are based on homology searches and sequence annotation and use sequences of proteins across genera of known localizations to find closely similar proteins to predict localization; these results are more significant and imply consistent subcellular localization. Similar AQP localization results were observed in common bean, bay bean, and mustard, although for the TIP subfamily, only BrTIP5-1 was observed in the plasma membrane. Exons, introns, and conserved motif structures provide clues regarding gene functions [35]. According to the *VaAQP* exon–intron structural study, the gene lengths in five different subfamilies differed significantly because of the different numbers and sizes of exons and introns. With the exception of VaTIP1-3, all of the VaTIP genes displayed two introns. Introns in VaAQPs ranged in number from zero to five. In case of NIPs, four out of ten members had three introns (*VaNIP3-3*, *VaNIP3-2*, *VaNIP31*, and *VaNIP1-5*), five members had four introns (*VaNIP2-1*, *VaNIP1-4*, *VaNIP1-2*, *VaNIP1-3*, and *VaNIP1-1*), and only one member had five introns (*VaNIP4-1*). Among the *VaPIP* genes, most members had three introns, although three members (*VaPIP1-3*, *VaPIP1-4*, and *VaPIP2-4*) contained two introns each, whereas *VaSIPs* had two introns. However, within the XIP family, early candidate evolution was indicated by the presence of only one intron in *VaXIP1-1* and no intron in *VaXIP2-1*, showing that the intron numbers of each subfamily of *V. angularis* AQPs are conserved. Similar results were also found for banana, chickpea, arabidopsis, and rice. Previous studies have demonstrated that the evolution of plant genomes frequently includes the addition or removal of introns and exons, which favors natural selection [36]. Similarly, the number of plant introns was correlated with gene expression, duplication, evolution, and diversification. Among the six motifs identified, motifs one, two, five, and six were widely found in the VaTIP, VaNIP, VaPIP, and VaXIP subfamilies, indicating that these families may have similar functions; however, motifs one, three, and four were absent in VaPIP1-3 and VaPIP1-4. In addition, motifs two and five were common to the TIP, NIP, PIP, XIP, and SIP subfamily members but were absent in VaNIP3-3 and VaNIP1-5. Motif six was lacking in the VaPIP and VaSIP subfamilies, while motifs three and four were predominantly contained in VaPIPs and found in one member of the VaSIP subfamily (VaSIP1-2). Similarly, VaNIP3-2 also contained motif four. The NPA domain was only carried out by motifs one and two. Overall, each protein subfamily had a similar motif composition. Moreover, the presence of two NPA motifs (asparagine-proline-alanine), as well as the ar/R selectivity filter and Froger’s residues in AQP proteins, were linked to substrate specificity and transport activity; therefore, characteristic structures can provide insights into the basic functions of VaAQPs. Previous research has shown that the NIP AQP subfamily shows large-scale permeation of substrates, such as water, arsenic, silicon, urea, and glycerol [37]. Considering the potential of NIPs, only four out of ten NIPs in VaAQPs contained double-conserved NPA motifs, whereas *VaNIP1-3* and *VaNIP3-2* frequently replaced the third residue A with V, and *VaNIP3-1* replaced S with V. Only *VaNIP1-4*, degenerated loop B NPA motifs into NPG. Although the *VaPIP* and *VaTIP* members also showed double NPA conserved motifs in their protein sequences, a wide discrepancy was observed in their NPA motifs as well as ar/R selectivity filters and Froger’s residues of all five subfamilies. The ar/R selectivity filter of the adzuki bean NIP1 group is typical of subgroup I of plant NIP AQPs, with residues W/V/A/R in the ar/R filter, low water membrane permeability, and transport of uncharged solutes, such as glycerol and formamide [38]. Surprisingly, only one NIP (VaNIP2-1), with two NPA signature motifs (Asn-Pro-Ala) with lengths of 288 and 108 amino acids (AAs) was found with a mutated ar/R SF (A/S/G/R) (as G/S/G/R in OsNIP2;1, CaNIP2-1, and MtNIP2-1 [29,30,39] and Froger’s residues (F/T/A/Y/F). Similar results were obtained for *P. vulgaris PvNIP2-1*, which presented a G to A mutation in the H2 position of the ar/R selectivity filter and thus could play a role in the homeostasis of silicon and boric acid. Because *VaNIP3-1* shares the same NPS/NPV aqueous pore and A/I/G/R ar/R selectivity filters, it could be involved in boron transportation.

Plant physiological processes are disrupted by hormones or severe environmental stress, particularly drought and saline conditions, which result in tissue dehydration. Typically, plants exposed to drought stress exhibit different morphological, physiological, biochemical, and metabolic modifications which lead to activating plant protective mechanisms [40,41,42,43]. Therefore, to determine the impact of drought stress on the plant, we examined plant attributes, chlorophyll content, and soil moisture content. We found that all the attributes were significantly reduced under drought stress. These results indicated that drought stress negatively impacted plant growth, root, shoot weight, and plant height, and also showed a reduction in the soil moisture content. The reduction of soil moisture content resulted in plant wilting and yellowing of the adzuki bean leaf. Similar impact of drought stress was previously reported by several plant species including adzuki bean [14,15,42,44]. Research has shown that AQPs play a pivotal role in maintaining cellular water and osmotic homeostasis based on their differential expression. In *V. angularis* most abiotic stress-responsive genes belonged to PIP-type genes (*VaPIP1-6*, *VaPIP1-7*, *VaPIP2-1*, and *VaPIP2-2*), TIP-type genes (*VaTIP3-1* and *VaTIP3-2*), and XIP type genes (*VaXIP1-1* and *VaXIP2-1* in varieties 17,235 and 17,033)*. VaPIP2-2* is commonly expressed in both variants and provides a comprehensive understanding of how AQPs affect *V. angularis* during drought. Although many reports have shown that the PIP and TIP subfamilies contain stress-responsive genes, few reports have revealed the particular functions of the NIP, SIP, and XIP subfamily members [37]. The accumulated expression of *VaPIP1-2* and *VaTIP5-1* indicated that they might be induced by drought in the leaves. These results particularly suggest that VaAQPs may play a distinctive role in the absorption of residual water during drought stress or osmoregulation to transport specific substrates and protect plants from dehydration. The differential expression profile of AQPs genes in different tissues serves as a useful foundation for establishing the regulatory role of AQPs in drought. The AQP family, which includes the PIP proteins, is well established to be crucial for water transport in the plasma membrane of many plant species [19,45,46]. Based on the expression analysis of PIP AQPs, previous studies demonstrated and identified the role of PIP genes in water transport in the plant. For example, the down-regulated PIP1 gene expression has shown reduced osmatic permeability in arabidopsis and tobacco (*Nicotiana tabacum*) leaf and root protoplasts, respectively, compared with wild-type plants and helped to increase the root to leaf dry mass ratio [46,47]. In addition, the expression level of PIPs in leaves modulates the transpiration rate in arabidopsis [48]. In our study, among the four *VaAQP* genes selected, *VaPIP2-1* and *VaPIP2-5* presented high expression in the root tissue under the control and drought treatments, as indicated by the qRT-PCR analysis. These results are aligned with previous studies and may provide a possible function in root water uptake and conductance to protect against drought stress and may thereby help stabilize/survive under 10-day drought stress, exhibit the feedback mechanism. In the roots of date palms, the expression of *PdPIP1-1*, *PdPIP1-3*, and *PdPIP2-2* was induced during early drought [49]. Similar results were obtained for rice, where *OsPIP2-2* showed higher expression under drought stress and *CrPIP2-3* from bay bean (*Canavalia rosea*) also fit the expression pattern in transgenic plants [50,51]. Another research study found that *SIPIP2-5*, *SIPIP2-7*, and *SIPIP2-1* were highly expressed in the roots [52]. These results indicate that PIP2 is not only important for maintaining the osmotic balance in roots but also for strengthening the water use efficiency during abiotic stress.

## 4. Materials and Methods

### 4.1. Genome-Wide Identification of Aquaporin Genes in Vigna angularis

The public reference genome of Vigan 1.1 was retrieved from the NCBI database (https://www.ncbi.nlm.nih.gov/genome/?term=vigna%20angularis (12 March 2022)). Using the protein sequences of *V. angularis*, a local database was formed using BioEdit ver. 7.2.5 [53]. Subsequently, a BLASTp search was performed using previously reported AQP genes from diverse model plant species, including arabidopsis (*AtAQPs*), rice (*OsAQPs*), and the legume crop soybean (*GmAQPs*). To identify high-scoring pairs (HSPs) of putative AQP genes in *V. angularis*, an e-value of 10−5 was used as the initial cutoff. The blast output was used to choose HSPs with >100-bit scores. Additionally, duplicate entries were removed, and only non-redundant hits were chosen for additional examination. Finally, the distinct putative AQP protein sequences were examined using SOSUI (https://harrier.nagahama-i-bio.ac.jp/sosui/sosuiG/sosuigsubmit.html (13 March 2022)) and TMHMM (http://www.cbs.dtu.dk/services/TMHMM/ (13 March 2022)) in order to determine transmembrane domains. The genome-wide distribution of *V. angularis* AQPs was obtained and visualized using TBtool software [54].

### 4.2. Structural Attributes of Aquaporin Genes and Proteins

Using the MEME motif search tool (http://mem-esuite.org/tools/meme (13 March 2022)) [55], conserved motifs of *V. angularis* AQPs were discovered by in silico analysis. During the analysis, default settings were employed. Further, using gene coordinates in the General/Generic Feature Format version 3 (GFF3) file, TBtools software (https://github.com/CJ-Chen/Tbtools (13 March 2022)) were used for the motif patterns and intron–exon structure analysis. Then the putative AQPs protein sequences were used to identify and confirm the MIP domain using the National Center for Biotechnology (NCBI) conserved domain search tool (https://www.ncbi.nlm.nih.gov/Structure/bwrpsb/bwrpsb.cgi (18 March 2022)). In addition, relative molecular weight (MW) and isoelectric point (PI) values were determined by the Expasy database (https://web.expasy.org/compute_pi/ (18 March 2022)). The subcellular localization of AQP proteins was predicted using CELLO v.2.5 (http://cello.life.nctu.edu.tw/ (18 March 2022)) and the WoLF PSORT tool (http://www.genscript.com/psort/wolf_psort.html (18 March 2022)).

### 4.3. Multiple Sequence Alignments and Phylogenetic Analysis of Aquaporin Genes

The *V. angularis* AQP (VaAQP) protein sequences were aligned with previously reported AQPs using the in silico method of the ClustalW program [56]. Subsequently, a phylogenetic tree of the 40 identified VaAQP protein sequences was constructed in Molecular Evolutionary Genetics Analysis (MEGA) software version 6 [57] using the maximum likelihood method, and the robustness of the branch was evaluated with 1000 bootstrap replicates. The AQP subfamilies PIP, SIP, TIP, NIP, and XIP were named based on previously discovered AQPs from arabidopsis, rice, and soybean [33,58]. In addition, a second phylogenetic tree was constructed using 40 AQP protein sequences from adzuki beans, along with 35 AtAQPs, 34 OsAQPs, and 72 GmAQPs.

### 4.4. Tertiary Protein Structure Prediction

The tertiary structure of VaNIP2-1 was predicted using Phyre 2 (http://www.sbg.bio.ic.ac.uk/~phyre2/html/page.cgi?id=index (28 March 2022)). Subsequently, the protein database file (PDF) generated by Phyre 2 was uploaded to MOLE (https://mole.upol.cz/ (28 March 2022)) to visualize the 3D structure and pore morphology.

### 4.5. Expression Profiling of Adzuki Bean AQPs

Raw RNA-seq transcriptomic data available from the bio-projects PRJNA576763, PRJNA577173, PRJNA629451, and PRJNA318974 were retrieved from the NCBI Sequence Read Archive (SRA) database (https://www.ncbi.nlm.nih.gov/sra (28 March 2022)). Raw reads were examined, mapped to the reference genome assembly (Vigan1.1), and downloaded from the NCBI database. The reads were used for de novo assembly using QIAGEN Aarhus, CLC Genomics Workbench 12.0.1, (www.qiagenbioinformatics.com (28 March 2022)).

The normalized reads per kilobase of transcript per million mapped (RPKM) values for the AQPs identified in the present study were extracted. Based on the RPKM normalized values, an expression heat map for all AQPs was constructed via in silico method by The Institute for Genomic Research (TIGR) Multi Experiment Viewer (MeV) (http://www.tm4.org/mev.html (28 March 2022)) program.

### 4.6. Plant Materials and Growth Conditions

The wild adzuki bean accession (IT 305544) was used for the gene expression analysis under drought conditions in the control environment of a growth chamber. Seeds were sown in pots [13.0 cm (diameter) × 10.5 cm (height)] containing horticultural soil (Tobirang, Baekkwang Fertility, Korea), with a photoperiod (14 h daytime, temperature:28 ± 2 °C, 60% to 80% relative humidity). The experiment was conducted in March 2022. Initially, all pots (empty) weight were measured and then equal weights of soil were filled in the pots. Seedlings were grown until the vegetative growth (V2) stage under well-watered conditions (100 mL of distilled water was added daily); to avoid waterborne contamination we used distilled water. In the experiment, eight pots were used and then divided into two groups, namely, water control plants (4 pots) and drought treatment plants (4 pots), two plants per plot for the later stage of the experiment. Further, the water supply was cut off over the next 10 days in the drought treatment plants but provided normally to the control plants. Also determined the impact of drought stress on plant attributes such as root, shoot, plant height, chlorophyll content, and soil moisture content. Based on the soil moisture content and the leaf wilting symptoms drought stress was determined. The chlorophyll content of leaves was measured using a leaf chlorophyll meter (soil plant analysis development (SPAD) chlorophyll meter, Minolta Corp. Ramsey, NJ), and soil moisture content was measured using ProCheck handheld reader (ICT Int. Australia). Leaf and root tissues were collected from all plants after 10 days of drought treatment. The experiment was repeated two times, and four plants were used for each treatment. All sample tissues were immediately submerged in liquid nitrogen and stored at − 80 °C until further RNA extraction.

### 4.7. RNA Extraction and qRT-PCR

Total RNA with high integrity 28S:12S ratio of 2:1 was extracted from flash-frozen leaf and root tissues from the control and drought samples using the RNeasy Plant Mini Kit (Qiagen, Valencia, CA, USA) following the manufacturer’s recommendations. Using the (1 μg) of total RNA cDNA was synthesized by RNA-to-cDNA EcoDryTM Premix Kit, (Takara, Kusatsu, Shiga Prefecture, Japan). By a PowerUp SYBR Green Master Mix Kit from Thermo Fisher Scientific Inc., VaAQP genes were processed to a quantitative real-time polymerase chain reaction (qRT-PCR) on an ABI StepOne PCR System (Applied Biosystems, Foster City, CA, USA). Briefly, 100 ng template cDNA and 10 nM each primer in a final volume of 20 μL according to manufacturer’s instructions used the following protocol: UDG activation at 50 °C for 2 min, followed by polymerase activation at 95 °C for 2 min, denaturation 95 °C 15 s (Hold) and annealing and extension at 60 °C for 1 min (40 cycles), followed by melt curve analysis 95 °C 15 s, 60 °C 1 min, and dissociation 95 °C 15 s. Four biological replicates were used during the analysis, and adzuki bean actin gene was used as the internal control for normalization. The primer set used for qRT-PCR were designed using Primer QuestTM Tool (https://sg.idtdna.com/pages/tools/primerquest (22 April 2022)). The details of primer pairs used in this study are provided in Appendix A.

### 4.8. Statistical Analysis

To determine statistical significance, GaphPad Prism software (Version 7.00, 1992–2016 GraphPad) was used. The data were analyzed for standard error (±SE) and Student’s *t*-test was performed to determine significant differences at the 5% level.

## 5. Conclusions

In this study, we identified the 40 AQP genes in the adzuki beans. To the best of our knowledge, this is the first study to identify the AQP gene in adzuki beans. The in silico analysis determined that 40 VaAQPs genes are distributed across five subfamilies, a division which is supported by the phylogenetic, gene structure organization, evolution, motifs, introns-exons, transmembrane subcellular localization, and tertiary proteins analysis. By examining selected *VaAQPs* gene expression profiles under drought stress in the root and leaf tissues of the adzuki bean, we were able to demonstrate the regulatory involvement of AQPs under drought–stress conditions. This study will be beneficial as a resource for clarifying AQP’s potential function in adaptation to challenging environments and for investigating the identified candidate genes of adzuki bean for functional characterization.

## Figures and Tables

**Figure 1 ijms-23-16189-f001:**
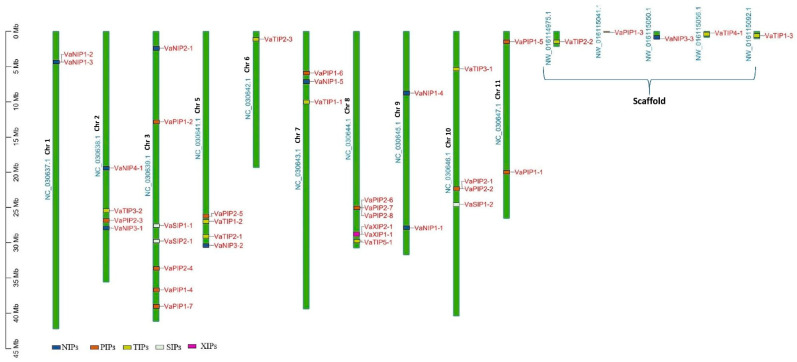
Genome-wide distribution of AQPs genes on the chromosomes of *Vigna angularis.* The right side of the chromosomal bar indicates the names of VaAQP genes, and the position is mentioned in megabase pairs (Mb) given on the left side. The “VaNIP” genes are represented in blue, “VaPIP” genes are in orange, “VaTIP” genes are in yellow, “VaSIP” genes are in white and “VaXIP” genes are in pink color.

**Figure 2 ijms-23-16189-f002:**
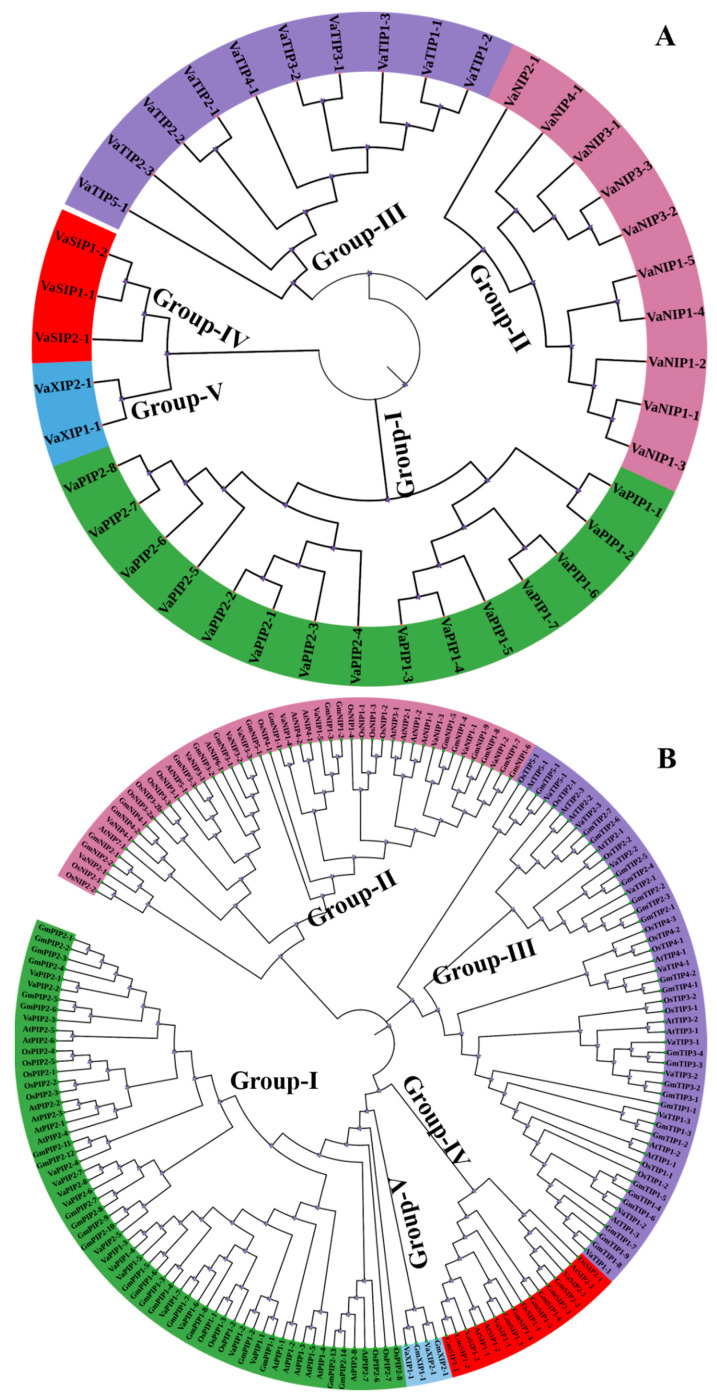
(**A**) Phylogenetic distribution of 40 aquaporins (AQPs) identified in *Vigna angularis* (*VaAQPs*); and (**B**) phylogenetic analysis of 40 VaAQPs along with the previously identified AQPs from *Glycine max*, *Oryza sativa* and *Arabidopsis*. Prefixes indicate different species, such as At—*A. thaliana*, Gm—*G. max*, Os—*O. sativa* and Va—*V. angularis* followed by subfamily name and specific number, which are used for the nomenclature of AQPs.

**Figure 3 ijms-23-16189-f003:**
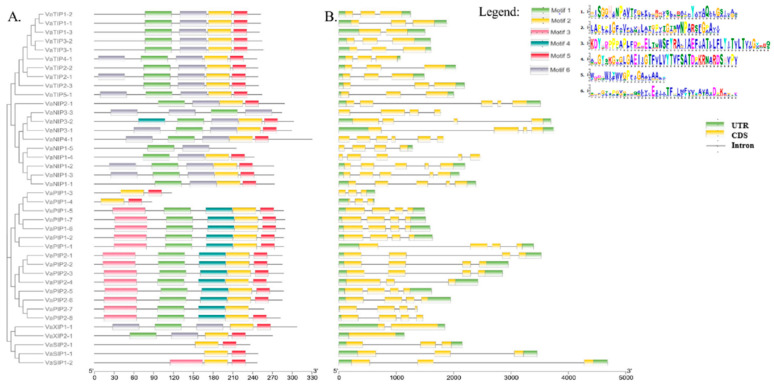
Conserved motifs of the VaAQPs proteins were identified using MEME. (**A**) Neighbor-joining evolutionary trees were generated using 1000 bootstraps, and a total of six motifs were identified among 40 AQPs genes, which are represented with the help of different colors. (**B**) Gene structure analysis and organization of the intron–exon of *VaAQPs*. Exon–intron structure analyses were performed using the TBtool; yellow boxes represent the CDS, green boxes represent the UTR, and black lines represent introns. The scale below Figure 3 represents the position of the CDS, UTR, and motif from 5′ end to 3′ end. The units used in the scale of Figure 3A are in amino acid (aa) and for Figure 3B is base pair (bp).

**Figure 4 ijms-23-16189-f004:**
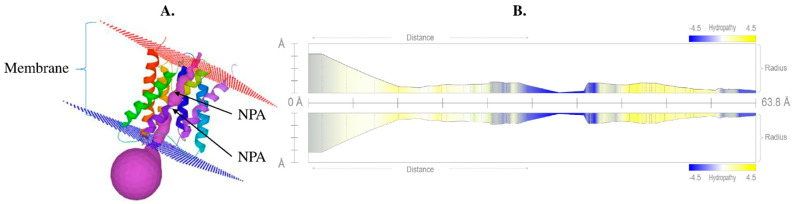
(**A**) Protein tertiary structure of silicon influx transporter VaNIP2-1 in *Vigna angularis* that clearly shows the presence of six transmembrane domains with NPA motifs; (**B**) hydropathy profile of pore with its radius. All the different colors in part (**A**) represent 6 different helices, and violet color represents pores. The yellow color represents the presence of hydrophobic residues whereas blue represents hydrophilic residues in (**B**).

**Figure 5 ijms-23-16189-f005:**
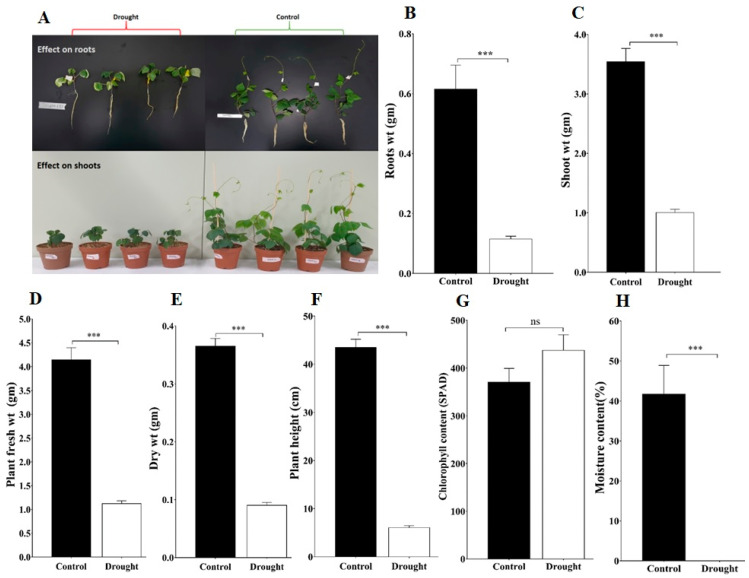
Effects of drought stress on the adzuki bean plants: (**A**) Evaluation of adzuki bean plant under drought and control conditions; (**B**) Plant roots weight; (**C**) Plant shoot weight; (**D**) Plant fresh weight; (**E**) Plant dry weight; (**F**) Plant height; (**G**) Chlorophyll content; and (**H**) Soil moisture content. Under control and drought stress conditions at 10th day of treatment. Data are means ± SD of n = 4 biological replicates. Error bars represent standard deviation. Means were analyzed for significant differences using the Student’s *t*-test. Asterisks (*) indicate differences at 5% level of significance (*** *p* < 0.001; ns, nonsignificant.).

**Figure 6 ijms-23-16189-f006:**
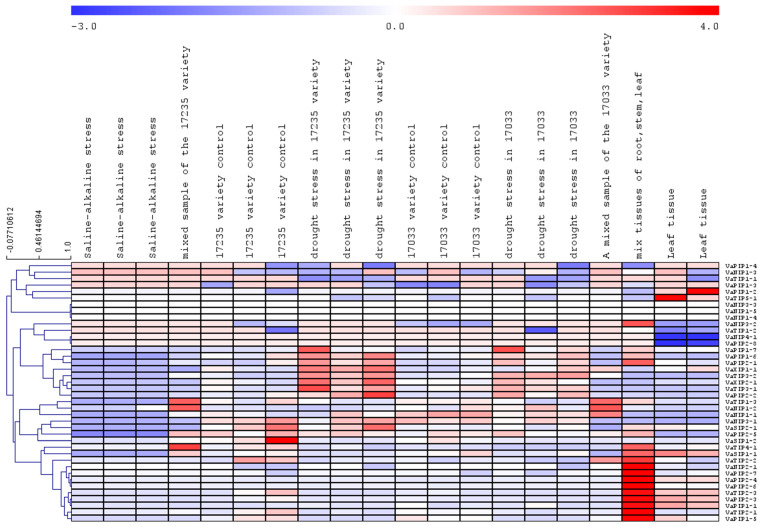
Heatmap and hierarchical clustering of the expression values relative to *V. angularis aquaporins*. The heatmap expression of all the aquaporins in different conditions is based on the RPKM values. The blue color represents low-expressed genes and the red color represents highly expressed genes.

**Figure 7 ijms-23-16189-f007:**
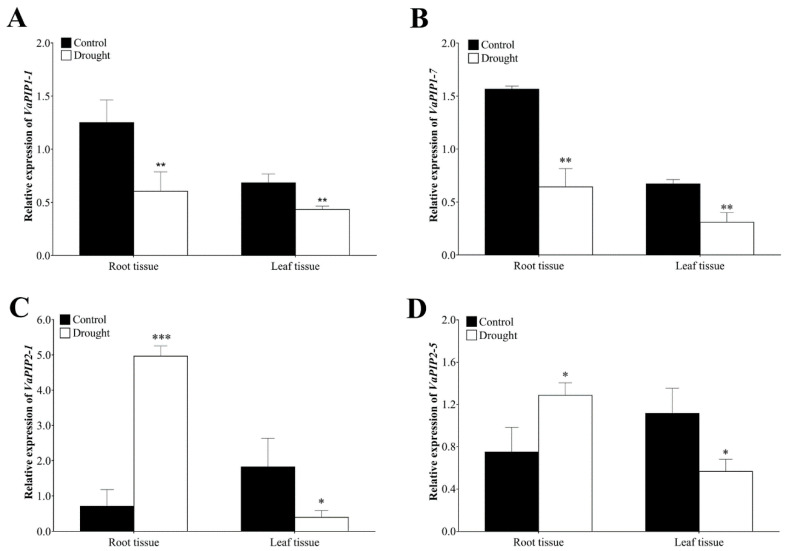
Expression analysis of *VaAQPs* genes under drought stress and control plants of the adzuki bean. The relative expression levels of (**A**) *VaPIP1-1*, (**B**) *VaPIP1-7*, (**C**) *VaPIP2-1*, and (**D**) *VaPIP2-5* in root and leaf tissue. Data are means ± SD of n = 4 biological replicates. Error bars represent standard deviation. Means were analyzed for significant differences using the Student’s *t*-test. Asterisks (*) indicate differences at 5% level of significance (*** *p* < 0.001, ** *p* < 0.01, * *p* < 0.05).

**Table 1 ijms-23-16189-t001:** Details of the transmembrane helix predicted by TMHMM and SOSUI and subcellular localization predicted by CELLO and WoLF PSORT.

AQPs	Gene Id	Len	ExpAA	First60	Predicted Helix		Subcellular Localization	
					TMHMM	SOSUI	CELLO	WoLF PSORT
>VaTIP2-2	XP_017407401.1	248	154.87	29.45	7	6	Plasma membrane	Vacuole
>VaPIP1-3	XP_017408780.1	117	44.66	22.07	2	2	Plasma membrane	Peroxisome
>VaNIP3-3	XP_017408931.1	284	135.19	40.11	6	8	Plasma membrane	Plasma membrane
>VaTIP4-1	XP_017409180.1	246	145.75	33.54	7	6	Plasma membrane	Vacuole
>VaTIP1-3	XP_017410025.1	250	148.13	26.18	7	6	Plasma membrane	Plasma membrane
>VaNIP1-3	XP_017411834.1	272	113.48	20.39	5	6	Plasma membrane	Plasma membrane
>VaNIP1-2	XP_017411847.1	272	128.31	20.56	6	6	Plasma membrane	Plasma membrane
>VaNIP4-1	XP_017413066.1	330	133.66	0.75	6	7	Plasma membrane	Plasma membrane
>VaNIP3-1	XP_017413544.1	299	116.69	0.02	5	6	Plasma membrane	Plasma membrane
>VaPIP2-3	XP_017414933.1	287	133.34	20.29	6	5	Plasma membrane	Plasma membrane
>VaTIP3-2	XP_017415113.1	254	131.52	22.33	6	6	Plasma membrane	Plasma membrane
>VaPIP1-2	XP_017415825.1	287	129.56	5.45	6	6	Plasma membrane	Plasma membrane
>VaSIP1-1	XP_017415827.1	248	130.84	33.04	6	5	Plasma membrane	Plasma membrane
>VaSIP2-1	XP_017415846.1	236	114.37	42.85	5	3	Plasma membrane	Vacuole
>VaNIP2-1	XP_017417585.1	288	114.17	11.98	5	6	Plasma membrane	Vacuole
>VaPIP1-4	XP_017418201.1	87	45.08	27.3	2	2	Plasma membrane	Peroxisome
>VaPIP2-4	XP_017419065.1	284	131.22	20.96	6	6	Plasma membrane	Plasma membrane
>VaPIP1-7	XP_017419407.1	289	128.09	4.22	6	6	Plasma membrane	Plasma membrane
>VaTIP1-2	XP_017424211.1	252	143.82	25.7	6	6	Plasma membrane	Plasma membrane
>VaNIP3-2	XP_017424310.1	302	130.14	0.08	6	6	Plasma membrane	Plasma membrane
>VaTIP2-1	XP_017424763.1	248	151.49	29.42	6	7	Plasma membrane	Plasma membrane
>VaPIP2-5	XP_017425377.1	287	127.98	19.9	6	6	Plasma membrane	Plasma membrane
>VaTIP2-3	XP_017426514.1	249	140.66	29.25	6	7	Plasma membrane	Vacuole
>VaNIP1-5	XP_017428564.1	215	98.39	22.42	5	4	Plasma membrane	Plasma membrane
>VaTIP1-1	XP_017429504.1	252	135.74	25.28	6	6	Plasma membrane	Plasma membrane
>VaPIP1-6	XP_017430460.1	289	127.03	3.53	6	6	Plasma membrane	Plasma membrane
>VaXIP1-1	XP_017431397.1	307	129.32	17.19	6	6	Plasma membrane	Plasma membrane
>VaPIP2-7	XP_017432054.1	257	109.36	22.17	5	5	Plasma membrane	Plasma membrane
>VaXIP2-1	XP_017432779.1	270	130.44	43.24	6	7	Plasma membrane	Plasma membrane
>VaTIP5-1	XP_017432819.1	254	141.01	26.33	7	7	Plasma membrane	Plasma membrane
>VaPIP2-8	XP_017433605.1	282	129.84	22.14	6	6	Plasma membrane	Plasma membrane
>VaPIP2-6	XP_017433699.1	285	130.67	20.66	6	5	Plasma membrane	Plasma membrane
>VaNIP1-4	XP_017434552.1	242	127.5	24.19	5	6	Plasma membrane	Plasma membrane
>VaNIP1-1	XP_017435093.1	273	129.06	17.44	6	6	Plasma membrane	Plasma membrane
>VaSIP1-2	XP_017438952.1	247	133.45	29.6	6	6	Plasma membrane	Vacuole
>VaPIP2-1	XP_017439936.1	285	136.52	22.37	6	5	Plasma membrane	Plasma membrane
>VaTIP3-1	XP_017439967.1	256	135.76	22.58	6	6	Plasma membrane	Plasma membrane
>VaPIP2-2	XP_017440026.1	285	136.52	22.37	6	5	Plasma membrane	Plasma membrane
>VaPIP1-1	XP_017442272.1	287	129.23	4.75	6	6	Plasma membrane	Plasma membrane
>VaPIP1-5	XP_017442471.1	287	129.46	7.26	6	6	Plasma membrane	Plasma membrane

**Table 2 ijms-23-16189-t002:** Details of the NPA domains, ar/R selectivity filter, and Froger’s and Mitani’s residues identified in the *Vigna angularis* aquaporins.

					ar/R Filter				Froger’s Residues					
Gene ID	Length	NPA (LB)	NPA (LE)	NPA-NPA	H2	H5	LE1	LE2	P1	P2	P3	P4	P5	Mitani’s Residue
VaTIP2-2	248	NPA	NPA	111	H	I	G	R	T	S	A	Y	W	P
VaPIP1-3	117	NPA	-	-	I	-	-	R	I	D	A	M	E	S
VaNIP3-3	284	NPV	-	-	T	-	-	I	Y	G	L	V	Y	F
VaTIP4-1	246	NPA	NPA	111	A	I	A	R	S	S	A	Y	W	P
VaTIP1-3	250	NPA	NPA	110	H	I	A	V	T	T	A	Y	W	P
VaNIP1-3	272	NPA	NPV	109	W	V	A	R	F	S	A	F	I	P
VaNIP1-2	272	NPA	NPA	109	W	V	A	R	F	S	A	Y	L	P
VaNIP4-1	330	NPA	NPA	108	A	V	G	R	Y	S	A	Y	M	P
VaNIP3-1	299	NPS	NPV	108	A	I	G	R	Y	T	A	Y	L	P
VaPIP2-3	287	NPA	NPA	118	F	H	T	R	Q	S	A	H	W	W
VaTIP3-2	254	NPA	NPA	111	H	I	A	R	T	A	A	F	W	P
VaPIP1-2	287	NPA	NPA	118	F	H	T	R	Q	S	A	H	W	W
VaSIP1-1	248	NPT	NPA	112	V	V	P	N	E	A	A	L	Y	W
VaSIP2-1	236	NPL	NPA	108	S	H	G	S	D	V	A	F	F	W
VaNIP2-1	288	NPA	NPA	108	A	S	G	R	F	T	A	Y	F	P
VaPIP1-4	87	NPV	-	-	I	-	-	P	I	S	S	V	F	H
VaPIP2-4	284	NPA	NPA	118	F	H	T	R	Q	S	A	Q	W	W
VaPIP1-7	289	NPA	NPA	119	F	H	T	R	E	S	A	H	W	W
VaTIP1-2	252	NPA	NPA	111	H	I	A	V	T	S	A	Y	W	P
VaNIP3-2	302	NPA	NPV	108	T	I	G	R	Y	T	A	Y	L	P
VaTIP2-1	248	NPA	NPA	110	H	I	G	R	T	S	A	Y	W	P
VaPIP2-5	287	NPA	NPA	118	F	H	T	R	Q	S	A	Q	W	W
VaTIP2-3	249	NPA	NPA	110	H	I	G	R	T	S	A	Y	W	P
VaNIP1-5	215	NPA	-	-	W	-	-	S	L	M	F	H	C	N
VaTIP1-1	252	NPA	NPA	111	H	I	A	V	T	S	A	Y	W	P
VaPIP1-6	289	NPA	NPA	119	F	H	T	R	E	S	A	H	W	W
VaXIP1-1	307	NPK	NPA	158	G	V	A	R	G	C	A	W	I	V
VaPIP2-7	257	NPA	NPA	118	F	H	T	R	Q	S	A	Q	W	W
VaXIP2-1	270	SPV	NPA	129	V	V	V	R	I	C	A	W	V	L
VaTIP5-1	254	NPA	NPA	110	S	V	G	C	V	A	A	Y	W	P
VaPIP2-8	282	NPA	NPA	118	F	H	T	R	Q	S	A	Q	W	W
VaPIP2-6	285	NPA	NPA	118	F	H	T	R	Q	S	A	Q	W	W
VaNIP1-4	242	NPG	NPA	110	W	V	A	R	F	S	A	Y	I	P
VaNIP1-1	273	NPA	NPA	109	W	V	A	R	F	S	A	Y	L	T
VaSIP1-2	247	NPT	NPA	112	I	I	P	F	E	A	A	F	Y	W
VaPIP2-1	285	NPA	NPA	118	F	H	T	R	Q	S	A	H	W	W
VaTIP3-1	256	NPA	NPA	111	H	I	A	L	T	A	S	F	W	P
VaPIP2-2	285	NPA	NPA	118	F	H	T	R	Q	S	A	H	W	W
VaPIP1-1	287	NPA	NPA	118	F	H	T	R	Q	S	A	H	W	W
VaPIP1-5	287	NPA	NPA	119	F	H	T	R	E	S	A	Q	W	W

Note: NPA, Asparagine-Proline-Alanine; Ar/R, aromatic/arginine; H2, transmembrane helix 2; H5, transmembrane helix 5; LB, loop B; LE, loop E; all the abbreviations used in the table are the abbreviated forms/codes of amino acids (G, Glycine; P, Proline; A, Alanine; V, Valine; L, Leucine; I, Isoleucine; M, Methionine; C, Cysteine; F, Phenylalanine; Y, Tyrosine; W, Tryptophan; H, Histidine; K, Lysine; R, Arginine; Q, Glutamine; N, Asparagine; E, Glutamic Acid; D, Aspartic Acid; S, Serine; T, Threonine).

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
