# Peer review of "Genome-Wide Identification of Aquaporin Genes in Adzuki Bean (Vigna angularis) and Expression Analysis under Drought Stress"

_ijms, 2022, doi:10.3390/ijms232416189_

Round 1
Reviewer 1 Report
The file is attached.

Author Response
Author's Pointwise Reply to the Review Report (Reviewer 1)
In this study, the AQPs genes of wild adzuki beans, which confer resistance to drought, were examined. The aim of this study, which was carried out in a controlled environment, was to investigate whether AQPs proteins, which are effective against abiotic stress, are present in adzuki beans. These results were supported by examining 40 AQPs in adzuki beans and AQP proteins in different plant species and determining the relationship between them. In addition to the molecular study, the results of the study were supported by the observation some phenotypic features. Although the study was carried out twice, the plant species used could not be fully explained, and why it was conducted on a small number of plants could not be adequately explained. Although the selected PIP genes showed significant results, the expression of all genes in roots and leaves was not analyzed.
Answer: We would like to thank the worthy reviewer for the time given to this manuscript. All the comments and suggestions were genuine, valuable and improved the quality of the manuscript. We highly appreciate your efforts and agree to the suggested changes. All the changes that were made in the revised MS can be found with the track change, highlighted in green color.
Although there is sufficient information on the introduction of adzuki bean and AQPs genes, some information can be provided for other phenotypic features studied. The study was conducted on wild adzuki beans; however, the purpose of this study was not disclosed. Moreover, the information provided in the discussion section may become more meaningful if it is expressed by creating a table. Wild adzuki bean accession, four biological replicas, and the cultivar names expressed created confusion. They need to be explained clearly.
Answer: As per the comments, we have added the information in the introduction about, why the plant wild specie was used. The majority part and aim of this study is to emphasize the significance of AQP genes in adzuki beans by identification and validation of the AQPs genes. To support it we have conducted a relatively small-scale biological experiment under drought stress. Further using available public-domain transcriptome data (where different cultivars were used for abiotic stress) we analyze (in silico, heatmap) and identified the genes expressed in different plant tissues and selected the genes which are highly expressed in the root tissue for expression analysis. However, we have a plan to further conduct the broad experiment with different accessions and cultivated species (sensitive and tolerant to drought) to explore selected genes functional role. Secondly, We have mentioned that “Genome-wide identification of AQP genes has been performed in numerous plant species, including rice (Oryza sativa), soybean (Glycine max L.), and bottle gourds (Lagenaria siceraria) (Matsuo et al., 2012; Deshmukh et al., 2015; Shivaraj et al., 2017), and these studies showed that AQP genes are conserved across plant species. However, AQPs remain poorly understood in many species” In adzuki beans, AQP gene information is sparse, particularly in terms of drought stress. That’s why we decided to identify the AQPs genes and validate them. Generally, wild accessions are considered to have native resistant genes or alleles for tolerance to biotic and abiotic stress with higher genetic diversity. Thus, wild adzuki bean accessions were used. Hope it solves the purpose.
The legend of the tables should be written in more detail, and each abbreviation should be explained. References are not consistent. This should be rearranged according to the reference content of the journal. Also, these two citations (Lestari et al., 2014 and Luo et al., 2014) are not available in the references.
Answer: Thank you for the suggestions we have corrected the information based on the comments. Provided the cited references.
Comments by line:
L194. Why was distilled water used? How many plants were grown in each pot?
Answer: Sorry for the inconvenience, as the plants were grown in a growth chamber to avoid any water born contaminant we used distilled water, two plants per pot were grown. We have incorporated the information.
L208. What are these biological replicas used? What is the difference between them?
Answer: It's nothing but four different plants as biological replicates were used for each control and treatment group.
L242. Supplementary table S2, not S1. It may be helpful to add a column that indicates the chromosome on which these AQPS genes are located. In addition, the numbers given here do not match those given in supplementary table 2. What does specific or generic represent? Are we able to understand that it is tissue-specific? Do you know the gene IDs of these proteins?
Answer: Thank you for your suggestions, as per the suggestion, we have changed the naming from S1 to S2 and also added the chromosome number on which the AQPs are present. Yes, we know the gene IDs for all the proteins, which are now included in table 2.
It might be easier to follow if the uses of colors and group names in figure 2 are the same.
Answer: Thank you for your suggestions, as per the suggestion, we have colored group the naming.
L301-303. Although the expression states that five motifs are observed in VaTIP2s, according to figure 3A this is not true for all its members (VaTIP2-2 and VaTIP2-3).
Answer: We modified the sentence to avoid confusion.
L325-329. The given expression does not match the data in Table 1.
Answer: We corrected the information. Thank you.
The labels of figure 4B cannot be fully read.
What exactly is meant in figure 6? What do colors represent? What do the same labels in columns indicate? Is there any difference between them? If so, what is this difference? As figure 6 is not clearly expressed, it is not easy to understand the accuracy of the statements written in 3.7.
Answer: Figure 6 represents the heatmap expression of all the aquaporins in different conditions based on the RPKM values. The blue color represents low-expressed genes and the red color represents highly expressed genes. Also provided information in new figure legends.
L383. It is necessary to be consistent in the protein names written.
Answer: Thank you for the suggestion, we have corrected it at the necessary places.
L387. Figure 6, not 5.
Answer: Thank you for the suggestion, it was a typo error we have corrected it now.
L391-392. What are the reasons for choosing these genes?
L432. The study here was done on Chinese cabbage (B. rapa var. pekinensis), not mustard (B. juncea).
Answer: Thank you for pointing it out, it was a typo error we have corrected it now.
L445. It may be more useful to compare legumes here. A table containing all subfamilies can be created for this purpose.
Answer: We appreciate your comments, however since the topic has already been covered in detail, adding a table might not be beneficial.
L455. Although this statement is true according to the Cello results, it is not true in the given Wolf Psort results. It should be explained why the Cello results were more significant.
L463: VANIP4-1 has 5 introns.
Answer: Thank you for the suggestion, corrected it now.
L470-471. Do you have a resource for this?
Answer: We didn’t get this point. Sorry for inconvience.
L496. I think you meant VaNIP3-2.
Answer: Corrected.

Reviewer 2 Report
The authors have to consider the following concerns
1- Figures 1, 3, and 4 are not readable, you have to increase the resolution
2- Make sure that gene names are in italic all over the manuscript
3- The manuscript requires language editing
4- Please check the font shape and size in Figure 6
5- Please consider these references
a- Kamal, K. Y., Khodaeiaminjan, M., Yahya, G., El-Tantawy, A. A., Abdel El-Moneim, D., El-Esawi, M. A., Abd-Elaziz, M. A. A., & Nassrallah, A. A. (2021). Modulation of cell cycle progression and chromatin dynamic as tolerance mechanisms to salinity and drought stress in maize. Physiologia plantarum, 172(2), 684–695. https://doi.org/10.1111/ppl.13260
b- Moreno, D. F., Parisi, E., Yahya, G., Vaggi, F., Csikász-Nagy, A., & Aldea, M. (2019). Competition in the chaperone-client network subordinates cell-cycle entry to growth and stress. Life science alliance, 2(2), e201800277. https://doi.org/10.26508/lsa.201800277
c- Yahya, G., Pérez, A. P., Mendoza, M. B., Parisi, E., Moreno, D. F., Artés, M. H., Gallego, C., & Aldea, M. (2021). Stress granules display bistable dynamics modulated by Cdk. The Journal of cell biology, 220(3), e202005102. https://doi.org/10.1083/jcb.202005102
d- Georgieva, M. V., Yahya, G., Codó, L., Ortiz, R., Teixidó, L., Claros, J., Jara, R., Jara, M., Iborra, A., Gelpí, J. L., Gallego, C., Orozco, M., & Aldea, M. (2015). Inntags: small self-structured epitopes for innocuous protein tagging. Nature methods, 12(10), 955–958. https://doi.org/10.1038/nmeth.3556
Author Response
We would like to thank the worthy reviewer for the time given to this manuscript. All the comments and suggestions were genuine, valuable and improved the quality of the manuscript. We highly appreciate your efforts and agree to the suggested changes. All the changes that were made in the revised MS can be found with the track change, highlighted in green color.
The authors have to consider the following concerns
- Figures 1, 3, and 4 are not readable, you have to increase the resolution
Answer: Thank you for the suggestion, we have submitted higher-resolution images.
2- Make sure that gene names are in italic all over the manuscript
Answer: Thank you for the suggestion, the suggested changes have been incorporated.
- The manuscript requires language editing
Answer: Thank you for the suggestion but before submission, this MS was edited by a professional English editing service by a native English language editor.
4- Please check the font shape and size in Figure 6
Answer: Thank you for the suggestion but we are unable to change the font shape and size as these are tool-generated images and the pictures with such fonts are the default output.
5- Please consider these references
a- Kamal, K. Y., Khodaeiaminjan, M., Yahya, G., El-Tantawy, A. A., Abdel El-Moneim, D., El-Esawi, M. A., Abd-Elaziz, M. A. A., & Nassrallah, A. A. (2021). Modulation of cell cycle progression and chromatin dynamic as tolerance mechanisms to salinity and drought stress in maize. Physiologia plantarum, 172(2), 684–695. https://doi.org/10.1111/ppl.13260
b- Moreno, D. F., Parisi, E., Yahya, G., Vaggi, F., Csikász-Nagy, A., & Aldea, M. (2019). Competition in the chaperone-client network subordinates cell-cycle entry to growth and stress. Life science alliance, 2(2), e201800277. https://doi.org/10.26508/lsa.201800277
c- Yahya, G., Pérez, A. P., Mendoza, M. B., Parisi, E., Moreno, D. F., Artés, M. H., Gallego, C., & Aldea, M. (2021). Stress granules display bistable dynamics modulated by Cdk. The Journal of cell biology, 220(3), e202005102. https://doi.org/10.1083/jcb.202005102
d- Georgieva, M. V., Yahya, G., Codó, L., Ortiz, R., Teixidó, L., Claros, J., Jara, R., Jara, M., Iborra, A., Gelpí, J. L., Gallego, C., Orozco, M., & Aldea, M. (2015). Inntags: small self-structured epitopes for innocuous protein tagging. Nature methods, 12(10), 955–958. https://doi.org/10.1038/nmeth.3556
Answer: We highly appreciate the suggestions and cited a couple of studies at relevant places.

Reviewer 3 Report
The manuscript deals with identifying aquaporin gene family members in adzuki bean (Vigna angularis) - an important leguminous crop cultivated mainly for food purposes in Asian countries. It is well written, with a clear aim and objectives, combined with the use of appropriate experimental design. Results are comprehensibly presented and adequately discussed in the corresponding sections. Consequently, the research presented provides both important contribution to the understanding of some of the mechanisms in drought tolerance and to the exact genetic composition of the aquaporin gene family in this crop.
Author Response
Answer: We would like to thank the worthy reviewer for the time and appreciation given to this manuscript.
Reviewer 4 Report
This study provides examination Genome-Wide Identification of Aquaporin Genes in Adzuki Bean (Vigna angularis) and Expression Analysis Under Drought Stress. Before recommending this article for publication, there are some shortcomings for that should be resolve.
Line 17 write complete botanical name of the plant.
The authors should present specific methods used in this study in the abstract.
Add some details on the nutritional, economic and medicinal values of the adzuki bean in second paragraph.
Line 70-73 previous studies word was used but only one study is cited.
Line 87 “transcriptomic and proteomics data in the public domain” should be cited with relevant studies. The following studies shall be consider.
https://doi.org/10.3390/genes13101699, https://doi.org/10.3390/ijms22179175,
In materials and methods links should be provide with databases and websites.
Also cite methods.
Check line 145.
Conclusion looks like summary of the results. Add future perspective of the study. Also modify the conclusion based on the results.
Author Response
Author's Pointwise Reply to the Review Report (Reviewer 1)
Reviewer 4
This study provides examination Genome-Wide Identification of Aquaporin Genes in Adzuki Bean (Vigna angularis) and Expression Analysis Under Drought Stress. Before recommending this article for publication, there are some shortcomings for that should be resolve.
Answer: We would like to thank the worthy reviewer for the time given to this manuscript. All the comments and suggestions were genuine, valuable and improved the quality of the manuscript. We highly appreciate your efforts and agree to the suggested changes. All the changes that were made in the revised MS can be found with the track change, highlighted in green color.
Line 17 write complete botanical name of the plant.
Answer: We highly appreciate the suggestions we wrote the scientific names for the plants.
The authors should present specific methods used in this study in the abstract.
Answer: Thank you for your suggestions, we used different in silico approaches for analysis, so it's not ideal to point out a single method, However, we have mentioned in silico method in the abstract as per the suggestions.
Add some details on the nutritional, economic and medicinal values of the adzuki bean in second paragraph.
Answer: Thank you for your suggestions, we have incorporated the sentence in the suggested place of the introduction.
Line 70-73 previous studies word was used but only one study is cited.
Answer: There were two stied mentioned, however, we have added one more citation.
Line 87 “transcriptomic and proteomics data in the public domain” should be cited with relevant studies. The following studies shall be consider.
https://doi.org/10.3390/genes13101699, https://doi.org/10.3390/ijms22179175,
Answer: We highly appreciate the suggestions we cited relevant studies.
In materials and methods links should be provide with databases and websites.
Also cite methods.
Answer: Thank you for your suggestions, as per the suggestion, we have incorporated the suggestive changes and we already mentioned details of “bio-projects PRJNA576763, PRJNA577173, PRJNA629451, and PRJNA318974 were retrieved from the NCBI Sequence Read Archive (SRA) database (https://www.ncbi.nlm.nih.gov/sra)”
Check line 145.
Conclusion looks like summary of the results. Add future perspective of the study. Also modify the conclusion based on the results
Answer: Thank you for your suggestions, as per the suggestion, we have incorporated the suggestive changes.

Round 2
Reviewer 1 Report
In general, there is no explanation for the plant species used. Although four genes were selected and their expression in the leaves and roots was examined, it was not explained why these genes were selected. In the introduction, adequate explanation and support have not been provided in terms of the phenotypic traits (plant height, chlorophyll content, soil measure) studied. The use of references in the text is not consistent. If the number system is used, it should be used throughout the article.L71. Reference 12 or 12-14? Why distilled water is used is not included in the article.
L186&202. If these plants have the same biological background, please use the word "plant" only.
L216. Supplementary table 2, not 1. The authors say in Supplementary table 2 that there are 40 AQPs in total, but there are only 35. "15 (13) PIPs, 10 (8) NIPs, and three (two) SIPs", so the numbers given here do not match those given in Supplementary table 2. What does specific or generic represent? What is the difference between complete size and mapped size? Do these sizes have units? If yes, what are they? Why do SIP1-1 and PIP1-4 (chromosome 3), NIP1-5 (chromosome 7), and PIP1-3 and NIP3-3 (chromosome ?) exist in figure 1 but not in supplementary table 2?
L219-222. According to supplementary table 2, these genes appear on chromosome 8. Why is chromosome 4 not in supplementary table 2? There are inconsistencies in the table, figure, and text. In the text, the presence of genes on chromosomes 11, 14, and 15 is detected, while in supplementary table 2, it is seen that chromosomes cannot be detected. As far as I know, adzuki bean's chromosome number is 11; do you have a source that he is 15?
Fig1. It should be explained on the figure the chromosome numbers, what the numbers on the left represent, and what the expressions on the left of each chromosome indicate. L228. Which are these 26 AQPs? In figures 2a and 2b, each subfamily is in a different group. If these group names are determined according to the group names of other crops, there is no consistency in the names. For example, while NIP is group 1 in figure 2b, group 2 is in 2a. Using the same color for these two shapes, either by subfamily or group, will help to understand the shape. For example, if figure 2a shows green in PIP, then 2b should also be green. Fig3. What do the scales below show? What is the unit? If motif 1 does not show utr and motif 2 does not show cds, different colors should be used for these two figures. L288. It is said that the expression has been corrected, but there is no change in the text. Check the NIP3-3 in table 1. Table1. All abbreviations (H, I, G, R, ...) should be explained. What does ar/R mean? Fig4. It should be explained what each color and number represents. L338-339. It is said that the protein names have been corrected, but there is no change in the text. Fig6. What do the same labels in columns indicate? For example, there are 3 saline-alkaline stresses and two leaf tissues. What are the differences between the two leaf tissues? Also why are the numbers inconsistent, why are there 3 saline-alkaline stresses when there are two leaf tissues? What is the difference between the 'control 17235 variety' and '17235 variety control'? If there is no difference, it should be written in the same way. While it was stated that wild accession 305544 was used in the study before, there are two different cultivar names (17033 and 17235) here. If 4 different cultivars were used, why are there results for only two of them? Why were not these cultivars mentioned before? Where is the result of accession 305544? L348-349. On what basis were these genes selected? Why were not the results of all genes studied? Fig7. VaPIP2-5 should be added to the legend. L399. Delete 29. L402. Although this statement is true according to the Cello results, it is not true in the given Wolf Psort results. It should be explained why the Cello results were more significant.
L 409. Delete 4. L437. Should be VaNIP3-2, not VaNIP2-2. Sources used in the text but not available in the references: Wallace and Roberts, 2005; Deshmukh et al., 2017; Kapilan et al., 2018; Luo et al., 2014; Chen et al., 2020.
Source 37 spelling in reference is not consistent.
Author Response
Author's Reply to the Review Report (Reviewer 1)
Answer: We would like to thank the worthy reviewer for the time given to this manuscript. All the comments and suggestions were genuine, valuable and improved the quality of the manuscript. We highly appreciate your efforts and agree to the suggested changes. All the changes that were made in the revised MS can be found with the track change, highlighted in green color.
In general, there is no explanation for the plant species used. Although four genes were selected and their expression in the leaves and roots was examined, it was not explained why these genes were selected. In the introduction, adequate explanation and support have not been provided in terms of the phenotypic traits (plant height, chlorophyll content, soil measure) studied.
Answer: We are thankful to the worthy reviewer for the comments it has improved the MS, We have provided the expiations for the concerns raised at necessary places.
For example: “Generally, wild accessions are considered to have native resistant genes or alleles for tolerance to biotic and abiotic stress with higher genetic diversity. Thus, wild adzuki bean accessions were used to test the relationship between the differential expression of these genes and differential responses to drought stress”.
Regarding phenotypic traits, mentioned text incorporated in the introduction “It is considered that plants show different mechanisms to mitigate drought stress, and evaluation of phenotypic traits related to physiology, morphological and biochemical traits are crucial under abiotic stress. In addition, these traits are altered by various factors, influence (plant height, chlorophyll content, and soil measure) and are essential for crop improvement, thus we also investigated phenotypic traits in this study. Mainly, we aim to emphasize the significance of AQP genes in adzuki bean by identifying and validating the AQPs genes and provide the candidate genes which are induced by drought stress for more in-depth functional characterization”.
The use of references in the text is not consistent. If the number system is used, it should be used throughout the article.
Answer: Sorry for the inconvenience couple of references (Deshmukh et al., 2017; Kapilan et al., 2018) was missed changing from text to number now we have corrected it.
L71. Reference 12 or 12-14?
Answer: Earlier it was a single reference but according to another reviewer's suggestion we added more references. So, please refer to the updated references.
Why distilled water is used is not included in the article. L186&202. If these plants have the same biological background, please use the word "plant" only.
Answer: We have provided information on why distilled water is used in the materials and methods section. Suggested change incorporated.
L216. Supplementary table 2, not 1. The authors say in Supplementary table 2 that there are 40 AQPs in total, but there are only 35. "15 (13) PIPs, 10 (8) NIPs, and three (two) SIPs", so the numbers given here do not match those given in Supplementary table 2. What does specific or generic represent? What is the difference between complete size and mapped size? Do these sizes have units? If yes, what are they? Why do SIP1-1 and PIP1-4 (chromosome 3), NIP1-5 (chromosome 7), and PIP1-3 and NIP3-3 (chromosome ?) exist in figure 1 but not in supplementary table 2?
Answer: Thank you very much for highlighting the mistake. In response to your suggestion, we have corrected the errors and made supplementary table S1 into supplementary table S2. We have added the remaining AQPs in supplementary table S2.
- 2. What does specific or generic represent? What is the difference between complete size and mapped size? Do these sizes have units? If yes, what are they?:
Specific means highly similar (The total number of conserved features/sites that were mapped onto the set of query sequences from specific hits. Each conserved feature/site is counted only once, regardless of how many query sequences on which it was found) and generic is broader (The total number of conserved features/sites that were mapped onto the set of query sequences from non-specific hits, because those non-specific hits belong to a superfamily whose representative is an NCBI-curated domain that has such annotations).
Complete size |
The total number of residues in the conserved feature/site that has been annotated on the domain model. |
Mapped size |
The number of residues in the query protein sequence that match residues in the conserved feature/site that was annotated on the domain model. |
As most of the publications not provided these types of minute details, it is expected that the researcher involves in such type of work/analysis is well aware of these terms. Hence we have not provided these details. Hope it clarifies the concern.
L219-222. According to supplementary table 2, these genes appear on chromosome 8. Why is chromosome 4 not in supplementary table 2?
Answer: We corrected the information. Due to the absence of AQPs on chromosome 4, we did not include chromosome 4 in the supplementary table.
There are inconsistencies in the table, figure, and text. In the text, the presence of genes on chromosomes 11, 14, and 15 is detected, while in supplementary table 2, it is seen that chromosomes cannot be detected. As far as I know, adzuki bean's chromosome number is 11; do you have a source that he is 15?
Answer: Sorry for the inconvenience, we included the genes present on a scaffold that’s the reason there was little confusion about the chromosome numbers. We thank the worthy reviewer for bringing this error to our attention. We have now corrected the mistakes. We redraw Figure 1. Also showed respective chromosomes and revised the results content.
Fig1. It should be explained on the figure the chromosome numbers, what the numbers on the left represent, and what the expressions on the left of each chromosome indicate.
Answer: Thank you for the suggestions, we have modified (Figure 1.) with suggested changes and also explained in figure legends.
L228. Which are these 26 AQPs?
Answer: These genes are mentioned in Table 2. Hence, we modified the sentence “However, TMHMM predicted six transmembrane domains in only 27 out of the 40 VaAQPs shown in Table 2”.
In figures 2a and 2b, each subfamily is in a different group. If these group names are determined according to the group names of other crops, there is no consistency in the names. For example, while NIP is group 1 in figure 2b, group 2 is in 2a.
Using the same color for these two shapes, either by subfamily or group, will help to understand the shape. For example, if figure 2a shows green in PIP, then 2b should also be green.
Answer: Thank you for your suggestions, naming of V. angularis AQPs was performed based on their grouping with known AQPs from the other three species but the figure group names are given manually, however according to suggestions we have redrawn the figure and given the same color to the same subfamily in 2a and 2b part of the figure.
Fig3. What do the scales below show? What is the unit? If motif 1 does not show utr and motif 2 does not show cds, different colors should be used for these two figures.
Answer: The scale below figure 3 represents the position of the CDS, UTR, and motif from 5’ end to 3’ end. The units used in the scale of figure 3A are in amino acid (aa) and for figure 3B is base pair (bp). We have incorporated the information in figure legends.
L288. It is said that the expression has been corrected, but there is no change in the text.
Answer: We modified the sentence.
Check the NIP3-3 in table 1. Table1. All abbreviations (H, I, G, R, ...) should be explained. What does ar/R mean?
Answer: The suggested changes have been incorporated. All the abbreviations used in the table are the abbreviated forms/ codes of amino acids. In the table Ar/R means aromatic/arginine. The explanation is provided in the footnote.
Fig4. It should be explained what each color and number represents.
Answer: Thank you for the suggestion. The suggested changes have been incorporated into figure legends.
Answer: - It has been modified according to the suggestions.
L338-339. It is said that the protein names have been corrected, but there is no change in the text.
Answer: - It has been modified according to the suggestions.
Fig6. What do the same labels in columns indicate? For example, there are 3 saline-alkaline stresses and two leaf tissues. What are the differences between the two leaf tissues? Also why are the numbers inconsistent, why are there 3 saline-alkaline stresses when there are two leaf tissues?
Answer: These are the replications used for the RNAseq analysis, as provided in the SRA database. The samples taken for the analysis depend on the availability of the data present in the SRA database.
What is the difference between the 'control 17235 variety' and '17235 variety control'? If there is no difference, it should be written in the same way.
Answer: It’s nothing but the same so as per the suggestions change has been incorporated.
While it was stated that wild accession 305544 was used in the study before, there are two different cultivar names (17033 and 17235) here. If 4 different cultivars were used, why are there results for only two of them? Why were not these cultivars mentioned before?
Where is the result of accession 305544?
Answer: Thank you for commenting on it, There are two parts,
First part: “Raw RNA-seq transcriptomic data available from the bio-projects PRJNA576763, PRJNA577173, PRJNA629451, and PRJNA318974 were retrieved from the NCBI Sequence Read Archive (SRA) database (https://www.ncbi.nlm.nih.gov/sra). Raw reads were examined, mapped to the reference genome assembly (Vigan1.1), and downloaded from the NCBI database. The reads were used for de novo assembly using QIAGEN Aarhus, CLC Genomics Workbench 12.0.1, (www.qiagenbioinformatics.com). The normalized reads per kilobase of transcript per million mapped (RPKM) values for the AQPs identified in the present study were extracted. Based on the RPKM normalized values, an expression heat map for all AQPs was constructed via in silico method by The Institute for Genomic Research(TIGR) Multi Experiment Viewer (MeV) (http://www.tm4.org/mev.html) program”.
Second part: We conducted a biological experiment using (IT 305544) wild azuki bean. We have mentioned that for gene expression analysis under drought stress “The wild adzuki bean accession (IT 305544) was used for the gene expression analysis under drought conditions in the control environment of a growth chamber”. For we have presented the qRT PCR expression data for selected genes in (Figure 7). All the results mentioned for the effect of drought stress on plant attributes and (qRTPCR) gene expression analysis from the wild accession IT 305544.
L348-349. On what basis were these genes selected? Why were not the results of all genes studied?
Answer: The role of the PIP AQP subfamily is well established for managing the water status of plants through controlling cell and tissue hydraulics and is also considered the most likely candidate for protein-mediated hydraulic conductivity in roots and leaves. In addition, based on the transcriptome data mining we also found higher expression of PIP genes in plant tissues (root, meristem and leave) therefore genes are highly expressed in diverse plant tissues (root, stem, leaf) selected for qRTPCR analysis. Due to limitations of the fund and molecular analysis reagents we checked a few gene transcript levels under drought stress through qRT PCR.
Fig7. VaPIP2-5 should be added to the legend. L399. Delete 29.
Answer: The figure 7 legend was corrected and, as suggested, 29 is deleted from the text.
L402. Although this statement is true according to the Cello results, it is not true in the given Wolf Psort results. It should be explained why the Cello results were more significant.
Answer: Thank you for the suggestions, we have modified the sentence and provided the explanation implying the significance of CELLO predictions.
L 409. Delete 4.
Answer: Thank you, suggested change has been incorporated.
L437. Should be VaNIP3-2, not VaNIP2-2.
Answer: Thank you, as per suggestion we have changed the VaNIP2-2 to VaNIP3-2.
Sources used in the text but not available in the references: Wallace and Roberts, 2005; Deshmukh et al., 2017; Kapilan et al., 2018; Luo et al., 2014; Chen et al., 2020.
Source 37 spelling in reference is not consistent.
Answer: We corrected the information.

Reviewer 2 Report
The authors were able to cover most of my concerns and I am pleased to accept the manuscript in the present form
Author Response
We would like to thank the worthy reviewer for the time given to this manuscript.